# Discovering and Measuring CDNs Prone to Domain Fronting

## ABSTRACT

Domain fronting is a network communication technique that involves leveraging (or abusing) content delivery networks (CDNs) to disguise the final destination of network packets by presenting them as if they were intended for a different domain than their actual endpoint. This technique can be used for both benign and malicious purposes, such as circumventing censorship or hiding malware-related communications from network security systems. Since domain fronting has been known for a few years, some popular CDN providers have implemented traffic filtering approaches to curb its use at their CDN infrastructure. However, it remains unclear to what extent domain fronting has been mitigated.

To better understand whether domain fronting can still be effectively used, we propose a systematic approach to discover CDNs that are still prone to domain fronting. To this end, we leverage passive and active DNS traffic analysis to pinpoint domain names served by CDNs and build an automated tool that can be used to discover CDNs that allow domain fronting in their infrastructure. Our results reveal that domain fronting is feasible in *22* out of *30* CDNs that we tested, including some major CDN providers like Akamai and Fastly. This indicates that domain fronting remains widely available and can be easily abused for malicious purposes.

### ACM Reference Format:
Anonymous Author(s). 2023. Discovering and Measuring CDNs Prone to Domain Fronting. In *Proceedings of ACM Conference (Conference'17)*. ACM, New York, NY, USA, 9 pages. https://doi.org/10.1145/nnnnnnn.nnnnnnn

## 1 INTRODUCTION

Domain Fronting, a technique designed to mask the true endpoints in network communications, works by leveraging (or abusing) shared hosting infrastructure provided by widespread services such as Content Delivery Networks (CDNs). By leveraging a CDN's shared infrastructure, applications may appear to connect to a domain A served by the CDN while, in reality, the traffic is intended for a different destination domain B that is served by the same CDN as well. As a result, a network traffic monitor (e.g., an intrusion detection system or a censorship enforcement device) may believe a client is connecting to domain A, rather than B. In countries with stringent internet restrictions, such as China and Iran, domain fronting has been instrumental for activists and ordinary citizens alike to bypass digital barriers and access platforms like Signal and Telegram [5, 9]. However, the same technique has found favor among malicious actors. For instance, APT29, also known as Cozy

Bear, reportedly used domain fronting to camouflage their malware command-and-control (C2) infrastructure, complicating detection and attribution [7]. Furthermore, according to a recent study [10], about 3.5% of all Cobalt Strike Beacons were configured to use domain fronting to effectively evade detection for a prolonged period of time.

In order to detect or defend against domain fronting, censors and network operators are compelled to adopt drastic CDN traffic blocking measures, often with considerable collateral damage, in an attempt to mitigate the associated risks [20]. Rather than blocking CDN traffic altogether, a more effective approach to counter this threat lies within the infrastructure of CDNs themselves. To prevent unintended consequences from nationwide censorship, few popular CDNs have taken measures to prevent domain fronting on their platforms. For example, Google and Amazon disabled domain fronting in their services in 2018 [1], while Microsoft Azure only disabled it recently in November 2022, following its use by Meek, a Tor plugin for traffic tunneling [4, 7]. Irrespective of these measures, there exists evidence that domain fronting may still be leveraged for both benign [30] and malicious purposes [14]. However, it remains unclear to what extent domain fronting can still be successfully used and on what CDN infrastructure.

In this paper, we present a comprehensive measurement of CDNs that are still prone to domain fronting, offering valuable insights for CDN customers, researchers, and security administrators. To this end, we develop an automated system capable of measuring the potential for domain fronting in a variety of real-world CDNs. Previous work [18] by Fifield et al. exposed and tested domain fronting on a limited number of popular web services and major CDNs, using mostly manual effort [18]. However, the proposed approach is costly, does not scale, and is insufficient to perform a comprehensive test of CDN infrastructure to determine what parts of the infrastructure is prone to domain fronting. Unlike [18], our proposed measurement system leverages readily available DNS data to discover information on domains linked with CDNs and *automatically perform domain fronting testing at a large scale without the need for registering any new domain names or hosting any new services behind each CDN*, thus largely reducing associated manual efforts and monetary cost.

Using our proposed measurement system, we first collect domain names served by 38 different CDNs. We found that, contrary to the belief that popular domains are associated only with popular CDNs (e.g., Akamai, Cloudflare, etc.), popular domains within the top 10k ranking according to the ranking list, Tranco [27], also use less popular CDNs. Using our automated measurement system, we then performed domain fronting testing on 30 of the 38 CDNs and we found that domain fronting remains possible in 22 of these CDNs. Contrary to results reported in a previous study [18], our findings reveal that domain fronting is currently still possible for popular CDN service such as Fastly and Akamai, as well as for a variety of less popular CDNs. This finding is also corroborated by third-party

evidence suggesting that Fastly is being used as an alternate service in Tor plugins like Meek and SnowFlake [21].

In summary, we make the following contributions:

- We design a new measurement system that leverages DNS analysis to find domain names related to web content served via CDNs and that can automatically test whether a CDN is prone to domain fronting.
- Unlike previous work, our system can automatically test for domain fronting vulnerabilities without the need for registering new domain names or subscribing new web services with a CDN, thus eliminating manual efforts and allowing us to continuously test for domain fronting vulnerabilities at scale.
- Using our measurement system, we tested 30 different CDNs, and found that 22 of them are still currently vulnerable to domain fronting, including popular CDNs such as Fastly and Akamai.

## 2 BACKGROUND AND MOTIVATION

### 2.1 Domain Fronting

Domain fronting exploits the discrepancy between the TLS server name indication (SNI) and the Host header in HTTPS requests related to web content served via Content Delivery Networks (CDNs). CDNs typically rely on the Host header to identify the origin web server[1] responsible for satisfying an HTTP request, while the SNI is used for correctly establishing a TLS session (e.g., identify and deliver the correct SSL certificate to the client). Because the SNI is visible to network traffic monitors but the Host header is not (since it is encrypted via TLS), the true endpoint of the communication (the domain in the Host field) can be hidden "behind" the *front domain* expressed in the SNI.

The inherent ability to conceal the true destination makes domain fronting an ideal choice for different use cases. For instance, domain fronting has proved to be a valuable tool in internet censorship circumvention. This is demonstrated by its adoption in a number of widely used applications such as Telegram [5] and Signal [9], which are otherwise restricted as part of nation-wide censorship enforcement. At the same time, domain fronting is viewed as a threat by authorities that implement censorship restrictions.

Unfortunately, domain fronting has also been adopted by malware developers to hide the communications from malware compromised machines to their command-and-control (C2) server [14, 25]. By abusing a legitimate popular domain name as a front domain, they can hide C2 communications from network security and traffic analysis systems. This allows the malware to evade detection and to maintain control over a compromised system for longer periods of time, enabling stealthy data ex-filtration, malware updates, etc.

Figure 1 provides an overview of the steps involved in the use of domain fronting. As an example, we consider the case of a compromised machine that uses domain fronting to hide malware C2 communications, though the steps are similar in other applications (e.g., for censorship circumvention). We assume that the attacker already knows that a benign domain name legitsite.com is served by a given CDN. Before infecting the victim, the attacker registers

---

[1]https://www.cloudflare.com/learning/cdn/glossary/origin-server/

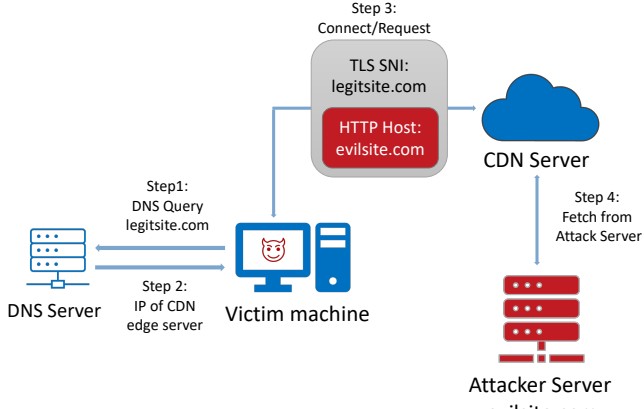

**Figure 1: Example of domain fronting use in malware.**

a new domain evilsite.com and subscribes it to the same CDN used by legitsite.com. Afterwards, the attacker infects victim machines with malware that uses domain fronting to connect to evilsite.com. As a first step, the victim machine issues a DNS query for legitsite.com, to find the IP address of a CDN server. Then, the malware initiates a TLS session with the CDN server, sets the SNI to legitsite.com, and sends an HTTP request to the server in which the Host header is set to evilsite.com. Upon reaching the CDN server, the web request is processed based on the information provided in the Host header (i.e., evilsite.com). If the related web content is not cached at the CDN's edge server, the CDN will forward the request to the evilsite.com origin server, obtains a response, and forwards the response back to the malware.

Notice that it is common for enterprise networks to use DNS and SNI monitoring to detect and block malicious communications. However, since both the DNS query (step 1) and SNI set by the victim (step 3) indicate a legitimate domain name, both the DNS request and the HTTPS connection will not be blocked.

### 2.2 Motivations for this Study

To enforce censorship in the presence of domain fronting, some nations, knowingly or unknowingly, have taken drastic measures to block CDN traffic, which resulted in blocking access to popular services such as Google and Amazon [20, 29], thus affecting millions of users. While this may be a potentially effective censorship strategy, this type of extreme countermeasure cannot be easily used to block malware communications in non-censored countries. For instance, consider *Exfiltrator-22* [14], a malware that uses domain fronting and abuses Akamai's CDN infrastructure to hide its C2 communications. Suppose, an enterprise network has been compromised by such a malware, and that the malware uses a set of popular legitimate domains as front domains. First, detecting the malware infection via network traffic analysis can be very challenging without sophisticated analysis of encrypted traffic [3], which may also be prone to false positives. Second, if the malware infection is identified and is found to abuse a set of popular domains, the network operator would need to block all traffic to those domains, which may include significant amounts of legitimate traffic. Alternatively, a defender may attempt to block

all IP addresses related to the abused CDN, but this would further increase the collateral damage. Furthermore, if traffic is blocked for a given front domain and CDN, the malware could automatically switch to a secondary front domain hosted on a different CDN that allows domain fronting.

Some popular CDNs have started to proactively mitigate domain fronting in their platforms by ensuring consistency between SNI and the Host header on all incoming web requests. While this is an effective countermeasure for preventing abuse, it is unclear to what extent domain fronting has actually been mitigated and the related challenges. For instance,

    (i) Do popular CDNs block domain fronting throughout their entire infrastructure?

    (ii) Are there other (perhaps less popular) CDNs that do not block domain fronting at all?

    (iii) Do these CDNs serve content from popular legitimate domains that can be abused as front domains?

These are some of the research questions we aim to investigate in this study.

## 3 MEASUREMENT METHODOLOGY

In this section, we provide an overview of our proposed measurement system to automatically test whether a CDN is prone to domain fronting. Figure 2 provides an overview of our system and its components.

Our main goal is to facilitate automated testing of domain fronting across many CDNs while avoiding the cumbersome manual process required for registering domains, subscribing CDN services, and paying for any associated costs. Our approach is based on the idea that, rather than registering our own domains with each CDN under test, we can identify domains registered by third-parties that already use those CDN services and leverage them to detect whether a CDN is prone to domain fronting. To achieve this, we first perform DNS traffic analysis to discover a list of domain names served by each CDN. Then, we test for domain fronting by selecting pairs of domains served by the same CDN to be assigned as either *target* domain or *front* domain. The target domain represents the actual destination of the HTTPS requests we will issue, while the front domain serves as the domain used to disguise the true destination of our HTTPS traffic.

The underlying idea is that if our tests succeed using existing domain names and web content served by a CDN, any actor (benign or malicious) can register a new domain name and subscribe to the same CDN's services and then use a third-party legitimate domain as front domain. To further confirm that the abuse of CDNs using domain fronting is in fact a real and current threat, we also measured the presence of malicious domains among the domain names that we identified as being associated with each of the CDNs in our data set. Specifically, we leverage Virus Total [8] to determine the percentage of CDNs that serve content from malicious domains. Our findings revealed that approximately 31% of the domain fronting prone CDNs served content from one or more malicious domains flagged by at least 2 security vendors in Virus Total platform.

While at a high-level this testing approach appears as quite straightforward, in practice, finding domain served by CDNs and testing CDN infrastructure for domain fronting is non-trivial. We discuss our approach in more details in the remainder of this section.

### 3.1 System Overview

As shown in Figure 2, our measurement system consists of three key components: (1) *CDN Domain Discovery*, (2) *URL Discovery*, and (3) *Domain Fronting Tester*. These components work together to (1) perform DNS analysis to discover website-related domain names whose content is served by a given CDN, (2) discover specific URLs under those domains that point to existing web content served via the CDN, and (3) use the information gathered from the two previous components to enable automated testing of domain fronting. We elaborate on the role of each system component below.

### 3.2 Discovering Domain Names Served by CDNs

Domain Fronting is possible if, and only if, the fronting domain and target domain are hosted on the same CDN. The first component of our measurement system focuses on finding the mapping between domain names and the CDNs that serve their content, by extracting relevant information from DNS records. When a web service $w$ under domain $d$ subscribes to a CDN, the CDN may assign it a custom subdomain $s.c$ of a domain $c$ owned by the CDN. This newly assigned subdomain can then be added as an alias of the subscribed domain in the DNS database for redirecting traffic to the CDN. Namely, a CNAME resource record can be registered for $d$ (the resource record name) that points to $s.c$ (the resource record data for the CNAME).

To find a list of domains served by a CDN, we proceed as follows. Given a CDN $C$ (e.g., Akamai, Fastly, etc.), we first compile a list $L_C$ of effective second-level domains (SLDs) used by $C$ to assign CNAMEs to its customers. We derive the list $L_C$ from an openly available list of CDNs [13] and extract SLDs for each CDN via manual search. Notice that this initial "seeding" step is the only manual step in our system, which serves to bootstrap our automated measurements.

Afterwards, to identify the list of domains related to websites that are served to a given CDN, we use passive and active DNS analysis. Specifically, we analyze DNS traffic passively collect at two large academic networks (with IRB approval) and openly available DNS data collected by the ActiveDNS project [24]. For every CDN's SLD, $c \in L_C$, we search the DNS datasets for CNAME records whose record data match $c$ (we use suffix matching for this). We then extract all resource records of the type $s.c$. To obtain the corresponding domain related to the website hosted by the CDN, we inspect the DNS response that included the CNAME $s.c$ and extract the query name $q$ from the question section of the DNS response where the CNAME was found. We repeat the process for each CDN.

Consider the following example to understand the steps involved. Let $c$ be edgekey.net. In this case, we use DNS analysis to collect subdomains of $c$ such as www.microsoft.com-c-3.edgekey.net, denoted as $s.c$. To obtain the corresponding domain related to the website hosted by the CDN, we inspect the DNS response that included the CNAME $s.c$ and extract the query name $q$ from the question section of the DNS response where the CNAME was found. In this example, $q$ = www.microsoft.com. By repeating this search for every CDN

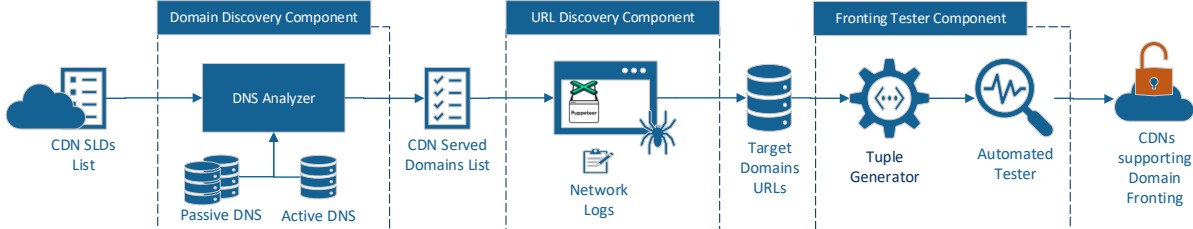

**Figure 2: Overview of Domain Fronting Measurement System**

domain $c \in L_C$, we derive the list $D_C$ of all domains visible from our DNS dataset that are related to websites served by CDN $C$.

### 3.3 Discovering CDN-served URLs

Once we have gathered the list of domains $D_C$ served by a CDN, as explained above, we proceed to map URLs that point to actual web objects under those domains. Namely, given a domain $q \in D_C$ such as assets.example.com, simply issuing a "GET /" HTTP(S) request for the "root" path under domain $q$ (i.e., https://assets.example.com/) may not work (an HTTP 4xx message may be returned) without specifying a full path to an existing resources under $q$. Therefore, to find valid URLs we proceed as follows.

Given a domain $q \in D_C$ such as assets.example.com, we first compute its effective second-level domain (in this case, example.com). Then, to find valid URLs under domain $q$, we have developed a custom Chromium-based web crawler using Puppeteer[6]. Our crawler is designed to visit and crawl a generic domain and capture details of all network requests and responses issued by the browser during the browsing session. This includes all requests related to web objects located under the visited domain directly as well as any of its subdomains. These allows us to discover a subset of full URLs $U_C$ for web objects served by the CDN. In the example above, pointing our instrumented browser to https://example.com and crawling its content allows to also discover web objects (i.e., their full URL) hosted under domain assets.example.com, which is the domain served by the CDN that we are interested in. To enable consistent fronting testing results, among the $U_C$ URLs we only retain those that correspond to static web resource, such as images, .js and .css files, etc., for which their content remains stable across multiple requests and can be used for our domain fronting testing module.

### 3.4 Domain Fronting Tester

Given a CDN $C$ and the related domain names and URLs it serves, which we discover as explained in Sections 3.3 and 3.2, the high-level domain fronting testing process is relatively straightforward: (1) select a domain name $d \in D_C$, (2) select a URL $u \in U_C$ whose domain $d_u$ is different from $d$, (3) establish a TLS session with SNI set to $d$ and (4) issue an HTTPS request for URL $u$ with the Host header set to $d_u$. If we are able to fetch a web object pointed by $u$ with no error, while the SNI points to $d$, the test succeeds and the CDN is prone to domain fronting.

Unfortunately, in practice, the process explained above is insufficient. The reason is that we also need to make sure that the object returned by the HTTPS request to $u$ is the same as the web object that the CDN would serve in a normal transaction (i.e., one in which URL $u$ is requested through the CDN without altering the SNI). Furthermore, we need to verify that this process works consistently for any paris of domains $(d, d_u)$ that are served by the CDN, to check whether all or only part of the CDN infrastructure is prone to domain fronting.

Therefore, we refine the testing process as follows. First, given a CDN $C$, we randomly select up to $N$ chosen tuples consisting of $(d_f, d_t, u_t)$, where $d_f$ is the *front* domain, $d_t \neq d_f$ is the *target* domain, and $u_t$ is a URL under $d_t$ that is served by the CDN (notice that $d_t$ and $d_f$ belong to the set $D_C$, whereas $u_t \in U_C$). The number $N$ depends on the cardinality of the sets $D_C$ and $U_C$.

For each selected tuple $(d_f, d_t, u_t)$, we proceed as follows:

- **Step 1: Request Target URL with Target Host** First, we craft a regular HTTPS request for URL $u_t$, so that both the Host header and SNI are set to the same domain $d_t$. We then store the response content (i.e., the requested web object), $r_t$, and use it as a reference to validate the result of the next test.
- **Step 2: Request Traget URL with Front Domain and Target Host** In this step, we test domain fronting. Specifically, we issue an HTTPS request for URL $u_t$ but set the TLS SNI to the front domain $d_f$ (the Host header is set to $d_t$). On receiving a valid response, we store it as $r_v$ and proceed with the next step.
- **Step 3: Request Target URL with Front Domain and Front Host** In this step, we craft a regular HTTPS request for $u_t$ but we replace $d_t$ with $d_f$. Namely, we set both the Host header and the SNI to $d_f$. We perform this step to ensure that the requested URL is not available under the fronting domain as well, since this would make the success of fronting test invalid. We store the response as $r_f$.

**Test Tuple Validation:** While selecting the $(d_f, d_t, u_t)$ tuples for testing, we apply additional filtering to avoid cases that would lead to potential false positives. For instance, domain fronting may be explicitly allowed between domains $d_t$ and $d_f$ if they are related to one another, for instance because one is a subdomain of the other or because the domains are owned by the same organization. In these cases, our test may lead to successful fronting tests even if a CDN proactively blocks domain fronting in general, when unrelated domains are set in the SNI and Host field. In practice, we confirm that domains $d_t$ and $d_f$ are related if: (i) they share the same effective

second-level domain, in which case we refer to them as "sibling" domains; or (ii) they are listed in a shared SSL certificate. To check for this latter condition, we analyze valid SSL certificates for each domain programmatically and check if $d_t$ and $d_f$ appear together in the Subject Alternative Name field (e.g., if *.example-1.com and *.example-2.net appear together in a valid SSL certificate, we consider them to be owned by the same organization). This further improves the confidence in the correctness of the results of our domain fronting tester.

**Fronting Test Validation:** By analyzing the responses, we determine that a single test was successful if (i) $r_t$ is a valid HTTP response (no HTTP or SSL error); (ii) $r_v$ matches $r_t$; (iii) $r_f$ is empty (i.e., no web object was retrieved) or is different from $r_t$. To compare the content of each response, we compute and compare their SHA1 hash. We repeat these tests up to $N$ times per each CDN (each test is based on a different randomly chosen tuple $(d_f, d_t, u_t)$, as explained earlier).

## 4 MEASUREMENT RESULTS

In this section, we present our results. Overall, our findings reveal that, despite domain fronting being known for a few years, there still exist many CDNs that are prone to it. Specifically, 22 out of 30 CDNs we tested are prone to domain fronting. Notably, we also observed successful domain fronting tests for popular CDNs such as Fastly and Akamai, which serve thousands of highly ranked domains (see Figure 3) that could be abused as fronting domains.

Besides detailed results regarding domain fronting, we also present additional findings and insights related to domain names served via CDNs, which can help understand the extent to which domain fronting may be successfully abused in practice.

### 4.1 Domain Analysis Results

To build a list of domains served by different CDNs, we leverage 10 days of passive DNS traffic collected (with IRB approval) from two large academic networks and via the ActiveDNS [24] project, between *March 20, 2023* and *March 30, 2023*. Specifically, we focus on CNAME resource records, which are typically used to direct web requests for domains served by CDNs to an edge CDN server (e.g., at the time of writing, querying for www.microsoft.com returns a CNAME chain pointing to an Akamai edge server). We inspect the CNAME resource records that match a large, manually curated list of second-level domains (SLDs) used by CDNs (see Section 3.2). To match the CNAME records against the list of CDN SLDs, we use suffix matching. We then keep only those CNAME records that match any of the SLDs in the curated list and discard the rest.

Now, let $D_f$ be the set of fully qualified domain names (FQDNs) for which at least one CNAME matched (via suffix matching) a CDN-related SLD. Our next step is to discover full URLs (including the full path to a web object) under those domains that are served by a CDN. To facilitate this next step, we proceed as follows. First, let $D_s$ be the set of all effective second level domains (SLDs) extracted from the FQDNs in $D_f$. We issue a "GET /" for each domain name $d \in D_s$ and keep all domains for which the "GET /" request returned a "200 OK" response. We call this reduced set $D_s'$ (domain names that return an error are filtered out). Overall, we found 38,567 domain names belonging to $D_s'$. We then consider all FQDNs $f \in D_f$

whose effective SLD belongs to $D_s'$, and call this new reduced set $D_f'$. Namely, $D_f'$ considers all subdomains of each domain in $D_s'$ whose content we found to be served by a CDN. After this step, we found 124,585 distinct FQDNs that are served by 38 different CDNs. Figure 3 shows the distribution of domains we identified per each CDN. As can be seen, most of the domains we collected are served by major CDN providers, with Cloudfront being responsible for serving 63% of the domains.

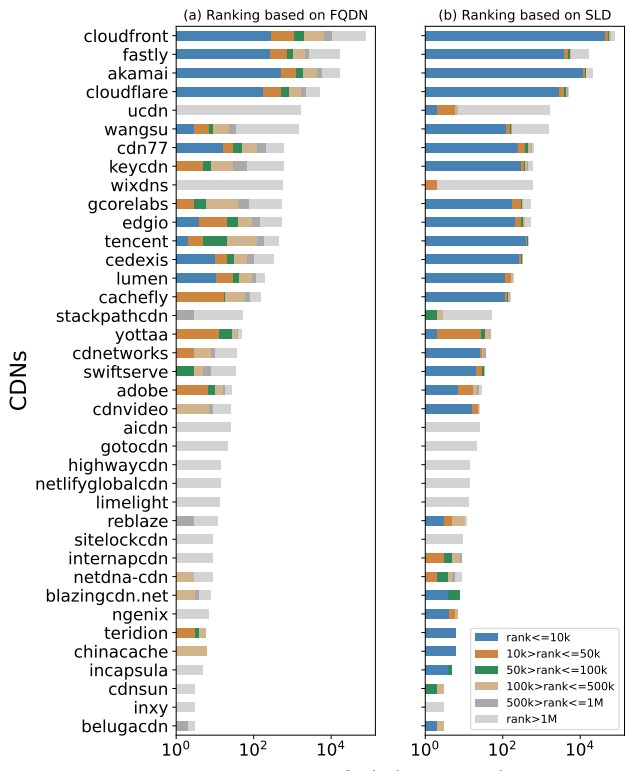

**Figure 3: Count of domains per CDN and their popularity ranking.**

Now, to discover full URLs served by CDNs, let $C$ represent one of the 38 CDNs we discovered so far. We randomly select up to 100 SLDs from $D_s'$ whose subdomains (at least one) or the SLD itself are served by $C$, and call this set of domains $D_s^{(C)}$. For each domain $d \in D_s^{(C)}$, we crawl the website pointed by $d$ and logs all HTTP requests and responses issued by our instrumented browser while rendering each web pages under $d$. We then store all full URLs for which a "200 OK" response was recorded in a set $U_C$. Finally, we reduce the set $U_C$ by only keeping a url $u \in U_C$ if its corresponding domain name $d_u$ belongs to $D_f$, thus forming a smaller set of URLs that we call $U_C'$. In summary, $U_C'$ is a set of URLs that point to web objects that are served by CDN $C$. Thus, at this point we know that an HTTPS requests for a URL $u' \in U_C'$ will go through $C$'s CDN infrastructure.

We repeat the above process for all 38 CDNs discovered so far. In the end, we were able to find valid URLs for 30 out of the initial 38

CDNs. Specifically, we found *52,998* URLs related to *1,310* distinct FQDNs served by those 30 different CDNs. For the remaining 8 CDNs, we were unable to find any full URL that we could use to issue HTTPS requests and fetch a valid web object via the CDN. It is possible that by crawling the web at large we could find URLs served by those remaining 8 CDNs as well. However, crawling the web in a non-targeted way can be quite time consuming and expensive in terms of resources (e.g., log storage). Therefore, we leave this enhancement step to future releases of our system.

**Popular Domains**: CDNs that host popular domains play a crucial role in the success of domain fronting, primarily due to the reduced risk of being blocked. When a popular domain is hosted by a CDN, there is a higher chance that the IP addresses associated with the CDN's edge servers will be considered benign and permitted by network security policies, even if those IP are shared by multiple domains. This proves advantageous to actors (malicious or benign) who leverage domain fronting as a means to mask their traffic and evade detection. Therefore, we also explored the distribution of popular (i.e., high rank) domains across the different CDNs.

To compute a domain's popularity, we use the Tranco [27] popularity list, which has been widely used in other web measurement studies. Specifically, we compute two different rankings for each domain name, one based on its fully qualified domain name (FQDN) and the other based on its effective second level domain (SLD) suffix. Figure 3 shows the distribution of popular domains, belonging to different ranking bands, served each CDN. Surprisingly, there are 26 CDNs that serve popular domains with $rank <= 10k$, based on their SLD. Even if we consider the ranking of FQDN, we can find 22 different CDNs that serve content from popular domains with $rank <= 500k$. This shows that, contrary to what one may have thought, highly popular domain names are not served only via the most popular CDNs (e.g., Akamai, Cloudflare, Fastly, etc.). Instead, the web content of some highly popular domains is served by less well known CDNs as well.

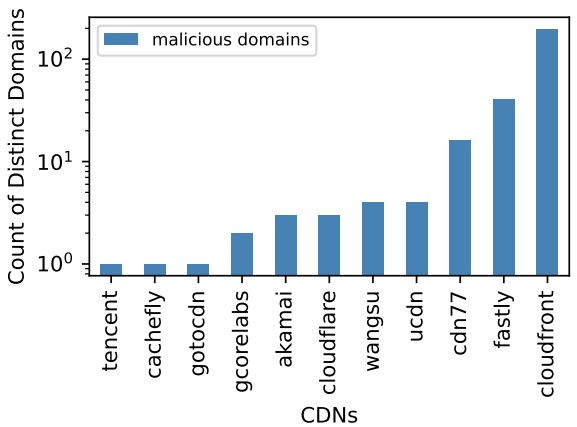

**Figure 4: Count of malicious domains (detected by 2 or more Virus Total Vendors) per CDN**

**Malicious Domains**: As mentioned before, CDNs can be abused for malicious purposes. Therefore, we also wanted to measure how many domain names served by each CDN are known to be malicious. To this end, we check each domain against VirusTotal [8], and flag domains that are labeled as malicious by different security vendors. We found 11 CDNs that served domains labeled as malicious by at least two different security vendors, and 27 CDNs serving one or more malicious domains flagged by at least one security vendor (see Figure 4).

*4.1.1 Additional CDN Insights.* To verify whether our method for discovering CDN-served URLs produced consistent results across different time windows, we conducted an additional experiment based on analyzing DNS traffic spanning multiple days. Our objective was to examine whether specific FQDNs were consistently associated with a single CDN throughout our DNS data collected over 10 days. The findings revealed that 99.64% of the FQDNs consistently mapped to (i.e., were served by) a single CDN, ensuring stable measurement results. The small fraction (0.36%) of domains that we found to be associated with different CDNs over time may be related to the use of Multi-CDN services [17]. A domain subscribed to such multi-CDN service could in real-time be associated with different CDNs based on various metrics, such as latency, performance overhead, proximity, demand and other factors. Considering that that number of such cases was negligibly small, we discard these domains from our dataset before conducting domain fronting tests.

## 4.2 Domain Fronting Test Results

Equipped with a large set of URLs served by 30 different CDNs (derived as explained in Section 4.1), we conducted our domain fronting tests (see Section 3.4) on those 30 CDNs.

**Table 1: Examples of DNS CNAME record**

| #CDNs Discovered via DNS | #CDNs with Popular Domains (SLD rank<=10k) | #CDNs with Malicious Domains | #CDNs for Testing | #CDNs Prone to Domain Fronting |
|---|---|---|---|---|
| 38 | 26 | 11 | 30 | 22 |

Figure 5 and Table 1 summarizes our results. We found that *22* out of 30 CDNs were prone to domain fronting, including some of the most popular CDN networks, such as Akamai and Fastly. To ensure that the results of our automated tests are accurate, we rely on the results of multiple test cases with varying parameters for each given CDN. In practice, for each CDN we generated multiple tuples, $(d_f, d_t, u_t)$, which we used for testing domain fronting as explained in Section 3.4. Because the number of all possible tuples that we could form for each CDN can be very large, to reduce the total time for the experiments and avoid causing any significant load on CDN infrastructure, we set an upper bound on the number of domain and URL combinations that we used for testing each CDN. Specifically, we randomly select up to 25 domains per CDN and up to 10 URLs per each domain.

Figure 5 shows the number of domains used for testing each CDN and the number of domains that were involved in successful domain fronting tuples. Overall, we found domain fronting to still work in 73% (22 out of 30) of the CDNs we tested. Among these,

there were 16 CDNs for which domain fronting tests were successful for all the domains we tried. In other CDNs, we found that not all domains could be used for successful domain fronting. Notably, popular CDNs such as Fastly and Akamai resulted in successful domain fronting tests for 100% and 52% of the tested domains, respectively. On average, in case of all 22 CDNs prone to fronting, domain fronting was successful for at least 50% of the tested domains. Further, our results also indicate that 8 of the CDNs had deployed mitigation measures against domain fronting throughout their entire infrastructure. Specifically, cases where 100% of the tests failed includes popular CDNs such as Cloudfront and Cloudflare, which is consistent with their public stance against domain fronting. For a few CDNs (e.g., `teridion`, `reblaze` and `inxy`), the number of domains we were able to discover and use for testing was small (e.g., <= 5) and our tests may be insufficient to confirm whether those CDNs have correctly implemented domain fronting mitigation throughout their entire infrastructure. Another interesting result is that, among the domains that we used in successful fronting tests, some were related to potentially sensitive domains, including www.census.gov.

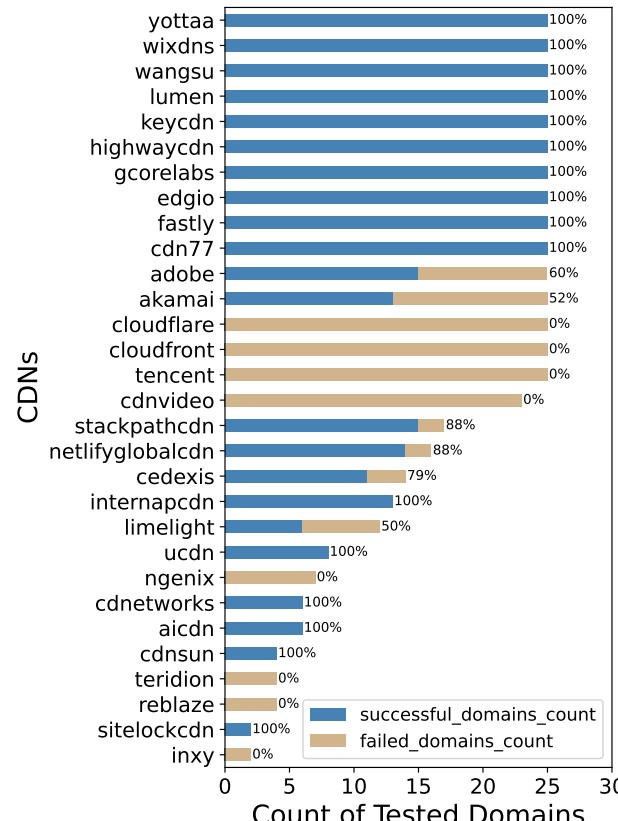

**Figure 5: Number of tested domains per each CDN and related domain fronting test success rate.**

*4.2.1 Analysis of CDNs with Partially Successful Fronting Tests.* To verify our results we manually analyzed a subset of our domain fronting tests. Especially, we focused on the 6 CDNs (shown in Figure 5) whose domain fronting tests succeeded for only a portion of all the test tuples (i.e., combinations of domains and URLs).

For instance, in case of Akamai, 13 of the 25 tested domains led to successful domain fronting. Our manual analysis revealed that, in all 12 cases that failed, the CDN servers used to provide the HTTP response were different from the CDN servers involved in the 13 successful domain fronting tests. Similarly, in case of StackPath, the 2 failed cases involved a CDN server not seen in any of the remaining 15 successful tests. Another case is represented by Adobe. In this case, all successful domain fronting tests involved CDN servers with domain name under the adobeaemcloud.com zone, whereas CDN servers with different names (e.g., under omtrdc.net, which is also an SLD related to Adobe's CDN) caused the domain fronting tests to fail. This indicates that while some of these CDNs have taken measures to mitigate domain fronting, they have not been able to do so consistently across their entire infrastructure. This insight was later partly confirmed during our responsible disclosure process, which we describe in Section 5.

In rest of the 3 CDNs involving a total of 11 domains, the response received was from a server belonging to a different CDN than the tested CDN. This is commonly observed in less popular CDNs and was found to be negligible(see section 4.1.1). This could be resolved by expanding DNS data to obtain more testable domains.

## 5 DISCUSSION

**Benefits of our proposed system**: Our system to detect CDNs that are prone to domain fronting can be beneficial to a diverse group of stakeholders. First and foremost, Internet freedom activists and journalists operating in regions with stringent internet censorship rules could use our system to learn which CDNs allow domain fronting and may be used bypass online restrictions, ensuring uninterrupted access to global information. Second, cybersecurity professionals and IT administrators would gain a better understand of potential ways in which attackers may "hide" from monitoring in their networks, and thus make more informed decisions on what CDN traffic to prioritize for detailed (and computationally expensive) inspection. Furthermore, CDN customers could themselves leverage our system to assess and analyze if their business might get affected as a collateral damage due to their CDN being prone to domain fronting and help them make a more informed choice on hosting services.

**Challenges in Automated CDN Detection**: In this project, our system automatically analyzes DNS records to identify domains whose web content is served by CDNs, provided the SLDs used by the CDN infrastructure are known. However, discovering the complete list of domains associated with each CDN and their SLDs is non-trivial. First, to the best of our knowledge, there are no public datasets of such CDN SLDs. Furthermore, by design, CDNs distribute content across a myriad of servers globally to optimize load times and provide redundancy. This distributed and dynamic nature inherently makes complete CDN infrastructure enumeration difficult. An alternate option is to use CDNFinder [11], a system that takes websites or domains as input and returns a list of CDNs serving those sites. CDNFinder employs multiple techniques, such

as pre-defined lists, HTML rewrite and IP-to-ASN mapping. However, CDNFinder is not suitable for our project, because we need to identify a wide range of CDNs, instead of focusing only on popular CDNs as in CDNFinder. Also, an initial analysis of CDNFinder results led us to discover a significant number of false positives. In a recent study [33], the authors focus on actively collecting DNS and HTTP based measurements to detect CDNs used by a given website. Similar to CDNFinder, this involves crawling large numbers of non-targeted set of domains that may or may not be associated with CDNs.

**Domain Fronting Mitigation at CDNs**: Cloudflare, a prominent CDN provider, offers an insightful example of how CDNs typically handle incoming web requests, which has implications for domain fronting [12]. As explained in the cited article, Cloudflare handles web requests using a multi-tiered system that includes two separate reverse proxies, a *TLS proxy* and a *business logic proxy*. When an incoming HTTPS connection reaches Cloudflare's infrastructure, it first encounters the TLS proxy, which is responsible for terminating the TLS connection. Then, subsequent HTTP requests are processed by the business logic proxy. In the context of domain fronting, the SNI is processed by the first reverse proxy, whereas the Host field in HTTP requests is processed by the second one. This may be one of the reasons why many CDNs do not block domain fronting, because it may be costly to adapt the underlying CDN infrastructure to make sure the two reverse proxy collaborate to check for consistency between the SNI and the Host field. In fact, our study revealed that, for some CDNs, only part of the CDN infrastructure is not prone to domain fronting. Based on discussions with CDN operators (as part of our disclosure process), we were told that only "new customers" (presumably served by newer infrastructure) are "protected" against domain fronting.

**Ethical Considerations**: It is important to note that our analysis of passive DNS traffic from two different academic networks was approved by the respective institutions. We only inspected DNS traffic to extract CNAME records and to map domain names to resolved IP addresses. Any other network traffic information was discarded. Also, it is worth noting that, to test CDNs for domain fronting, we only establish a limited number of HTTPS connection at a very low rate, and that all HTTP requests we issued are typical request for web objects. Therefore, we are confident that our measurements had no measurable impact on either the CDN infrastructure or the origin servers behind the CDNs.

**Responsible Disclosure**: We have already disclosed our findings to two large CDNs: Fastly and Akamai. Fastly has responded by acknowledging their awareness about the possibility of domain fronting within their CDN infrastructure. They mentioned that they have started to prevent domain fronting by default in their newer CDN service offerings, and that they deal with it on a case-by-case basis for other scenarios. We are awaiting a response from Akamai and we plan to continue our disclosure process to share our results with all other CDNs that we found to be prone to domain fronting.

## 6 RELATED WORKS

In this section, we discuss studies related to domain fronting and similar techniques. Fifield et al. were the first to introduce Domain Fronting in [18], which included results related to manually testing the capabilities of a small number of popular CDNs. To test whether domain fronting was possible, the authors registered their own domains and manually subscribed them to each of the CDNs being tested. Unlike their mostly manual testing approach, our proposed method is developed to conduct domain fronting tests at a large scale while also avoiding the need to register any new domain names. This greatly reduces manual efforts and costs associated with hosting domains and acquiring CDN services. A subsequent paper [32] proposes a different technique, named "domain shadowing," that can be used to abuse CDNs in combination with domain fronting. In [32], the authors highlight the threat faced by domain manipulation techniques that further serves as a motivation for our work, which sheds light on CDNs that are still prone to domain fronting. In addition, the focus of [32] is on domain shadowing, while we measure the prevalence of domain fronting. Yet another work [2] proposes a technique named "domain borrowing," which represents another way of abusing CDNs that is different from domain fronting.

Censorship circumvention, which is one of the applications of domain fronting,has also been studied from different angles [16, 19, 23, 26, 34]. These works primarily discuss applications of TLS in censorship and associated evasion tactics.

There also exists a number of studies [22, 28, 31] that showcase different methods related to abusing CDN infrastructure. Such works, in addition to recent reports of in-the-wild attacks that leverage domain fronting [10, 14, 25], demonstrate the growing threat posed by CDN abuse and the importance of automatically discovering potential vulnerabilities or paths of abuse in CDNs.

While working on this paper, we became aware of a very recent concurrent study [15] that was uploaded on arxiv.org in July 2023, which presents an approach to measure domain fronting that is similar to ours. To the best of our knowledge, [15] remains a *non-peer reviewed* paper at the time of this submission. While it partially overlaps with the our work, it did not in any way influence our research. We conducted the research described in this paper concurrently to [15]. In fact, we previously submitted an earlier version of this paper to another venue in May 2023, before [15] was uploaded to arxiv.org. Although this paper is an improved resubmission, the core of the paper is the same as the earlier version and we can provide evidence of this to the PC Chairs, if required, to prove that the two studies were done concurrently.

## 7 CONCLUSION

In this work, we successfully developed a measurement system that can be used to discover domain names and URLs that are served by a CDN, and to perform automated domain fronting tests to determine what CDNs are still prone to domain fronting. Through our evaluation, we discovered 22 CDNs that are prone to domain fronting, including highly popular CDNs such as Akamai and Fastly. The outcomes of our research offer valuable insights on CDNs and highlight the need for further efforts to prevent domain fronting abuse.

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
