# OpenReview forum: "Discovering and Measuring CDNs Prone to Domain Fronting"
_ACM.org/TheWebConf/2024/Conference — TheWebConf24 Oral_

### Official Review · Reviewer_cYsS · 2023-11-13

**Novelty:** 5
**Technical Quality:** 6

**Review:**

Dear authors, thanks for submitting your work to WWW 2024! This paper is a pleasant read overall, and I've learned a lot from it personally. While domain fronting is by no means a new topic, detailed measurement to demonstrate the status-quo is a meaningful contribution to the community. Some of the key results, e.g. "the web content of some highly popular domain names is served by less known CDNs as well", and "some CDNs haven't been able to do consistent mitigation across their entire infrastructure", are potentially important to both academic and industry communities.

With that being said I do have a few comments about the technical contributions. Automated testing is great, but the concern here is that completely relying on existing domains may leads to false positives and negatives. At the end of the day, attacks/censorship circumventions would have completely different characteristics and dependencies compared with other domains, so it is unclear whether such important differences could be taken into account in your solution. Moreover, for CDN providers with "partial" mitigation support, it would be good to automatically analyze the difference of protected domains v.s. unprotected domains to reveal the design decision/business logic underneath. The concrete suggestions would be to demonstrate that your method performs similarly or better than instrumenting new domains into CDNs, and to evaluate in detail what partial mitigation stands for.

As a side note, is the conclusion that we should always use cloudflare and cloudfront etc.? If not, why? what are some other aspects that could affect users' choices of CDN providers? Does this work really provide useful guidance to CDN selection? why not directly looking into company webiste and see whether they support domain fronting mitigation or not?

**Questions:**

Is the conclusion that we should always use cloudflare and cloudfront etc.? If not, why? what are some other aspects that could affect users' choices of CDN providers? Does this work really provide useful guidance to CDN selection? why not directly looking into company websites and see whether they support domain fronting mitigation or not?

**Reviewer Confidence:**

2: The reviewer is willing to defend the evaluation, but it is likely that the reviewer did not understand parts of the paper

**Scope:**

4: The work is relevant to the Web and to the track, and is of broad interest to the community

---

### Official Review · Reviewer_58J7 · 2023-11-17

**Novelty:** 4
**Technical Quality:** 5

**Review:**

The paper presents measurements and analysis to identify domain fronting abilities with CDN networks in todays Internet. As a part of the measurements, the authors identify that 22 of the 30 CDNs tested have make it feasible for clients to effectively use domain fronting using the CDN infrastructure. Domain fronting became widely known after Cozy Bear but has been adopted for non-malicious purposes in regions of the world with heavy censorship as mechanisms to evade firewalls and any other passive and active monitoring software deployed by governments and ISPs.

The paper aims to understand if the practice of domain fronting continues to be effective today because of the potential for malware masquerading their C2 mechanisms within these networks. Unlike the seminal works by Fifield et al., the authors argue about lowering the cost (manual effort and domain registrations) needed allowing researchers the ability to automatically test for domain fronting capabilities of CDN networks.

The authors achieved this through passive and active measurements of DNS and identifying CNAME records associated with CDNs and extracting prefixes, and discovering other unrelated domains which are also served by the same CDN provider. The authors present their methodology clearly and support them with examples throughout the paper making it incredibly easy to follow the results and writing. The authors identify popular top million domains using lesser known CDN networks countering the popular belief of popular websites using only large CDNs. The authors identify various CDN SLDs ($L_c$) and derive the list of domains matching the pattern. It would be valuable to the web community to share the different suffixes of the SLDs or the regex patterns used by various CDN networks.

The authors also carried out responsible disclosures about the domain fronting capabilities but should have also carried about reports to respective Trust and Safety teams within these CDNs about active domains classified as malicious by VirusTotal. Overall, the paper is easy to read and understand and could benefit from some editorial changes. The only weakness of the paper is the idea of domain fronting is not particularly new and open source efforts and prior research efforts have identified well known domain fronting lists [1] weakening the novelty of the contribution. The paper is clearly written, easy to understand and brings to light the continued tussle in addressing domain fronting by various CDN operators.

[1] https://github.com/vysecurity/DomainFrontingLists

**Questions:**

1. While the data for DNS queries and responses to extract CNAMEs came from ActiveDNS project and passively collected at two large academic networks, it is unclear why the same level of data/accuracy could not be obtained from public Zone file registries such as CZDS and identifying the DNS Nameservers associated with CDNs for each hostname and could be an efficient active measurement mechanism to obtain the mapping between SLDs/CDNs to the Domains deriving the list of domains $D_c$ served by a CDN.

2. Line 469-471 indicates that multiple hostnames being present in the SAN field of the certificate indicate them being owned by the same organization. However, this is not always true, a CDN service provider can provision the hosts needed to prevent overheads due to individual certificate management per host, and sometimes for performance or privacy reasons, an example would be the inclusion of a popular javascript/CSS/font library such as bootstrap or jQuery typically used by millions of webpages as an additional name with the SAN. While the authors argue that this is done to improve confidence of correctness of results, it is important to present the caveat of the case described above.

3. Figure 3 could benefit from clarification, given a set of FQDN domains $D_f$ matching the SLDs of the CDNs being considered in the measurement study,  shouldn't the ${ D_f } \rightarrow { D_s } $ ? For example $D_f$ is the set of `[A.com, B.com]` which match the CNAME records `www-a.server-asia-cdn.provider.com`, and `www-b.server-eu-cdn.provider.com` which match the SLD `*-cdn.provider.com`, shouldn't $D_s$ be the set of two CNAME records i.e. effective SLDs? In such a case, when comparing with the results in top million datasets, why are the proportions of the stacked bars on the left, and right different?

4. Line 582-584 indicates the inability to reach 8 CDNs because of lack of full URLs, given the domain/hostnames associated with these CDNs, have the authors attempted `HTTP "/" GET` requests or resolved the URLs to the records?

5. Figure 5 indicates Cloudflare, Cloudfront and tencent not being prone to domain fronting among the domains tested, does this indicate that despite the presence of malicious domains being served by these providers shown in Figure 4, the existence of these providers does is not a cause for concern regarding domain fronting? For the 11 CDNs listed in Figure 4, it would be valuable to indicate how many of these distinct domains classified as malicious can be domain fronted. Similarly extending Table 1 to indicate intersections of CDNs with malicious domains with those prone to domain fronting would be a useful metric to include.

6. It is unclear why resolving the `NS` records from DNS queries for a given FQDN is not a strong signal of the CDN being used requiring the need for alternatives like CDNFinder

7. Nit: Typo on Line 440, `Step 2: Request Traget URL` $\rightarrow$ `Step 2: Request Target URL`

**Reviewer Confidence:**

4: The reviewer is certain that the evaluation is correct and very familiar with the relevant literature

**Scope:**

4: The work is relevant to the Web and to the track, and is of broad interest to the community

---

### Official Review · Reviewer_9dLw · 2023-11-22

**Novelty:** 4
**Technical Quality:** 5

**Review:**

This work presents a measurement study of existing Content Delivery Networks (CDNs) that are still vulnerable to domain fronting attacks. It consists of three steps. First, it tries to identify the domain names served by different CDNs. This is done by using manual search to create a seed list of second-level domains (SLDs) associated with each CDN. Next DNS traffic is analyzed to extract the domain names matching the  SLDs for each CDN. In the second step, a web crawler is developed and used to discover the valid URLs served by each CDN based on the domain names discovered in the previous step. In the third step, customized HTTP requests are sent to the valid URLs served by the CDN to identify whether domain fronting is possible. The paper next presents the measurement results on 30 CDNs, showing that 22 of them are prone to domain fronting attacks. The findings were reported to two large CDNs, one of which responded by acknowledgement.

Strengths:
+ The findings from the measurement study can have real-world impact, by making the CDNs involved aware of the domain fronting issues.
+ The design of the measurement experiments is thorough with many technical challenges taken into consideration.
+ The paper is well written.

Weaknesses:
- The paper focused on the discovery of domain front issues, but it didn't discuss the plausible countermeasures against attacks abusing domain fronting.
- A similar study has already been published at EuroS&PW'23, although it's understandable that the two research efforts were actually performed independently.

**Questions:**

* In the discussion section, the work mentioned that there are legitimate reasons for CDNs to allow domain fronting in their networks. Given that, what would be the best practical ways to mitigate the risks associated with domain fronting?

**Reviewer Confidence:**

2: The reviewer is willing to defend the evaluation, but it is likely that the reviewer did not understand parts of the paper

**Scope:**

4: The work is relevant to the Web and to the track, and is of broad interest to the community

---

### Official Review · Reviewer_4HYA · 2023-11-23

**Novelty:** 3
**Technical Quality:** 4

**Review:**

This paper introduces an automated methodology for identifying CDN providers that endorse domain fronting. The approach relies on a combination of passive DNS traffic analysis and active DNS measurements to identify domains associated with CDN services. Subsequently, an automated process is employed to ascertain whether these identified CDN providers support domain fronting. The findings disclose that among the examined 30 CDN providers, 22 are confirmed to support domain fronting.

Pros:

-The proposed automated method for detecting domain fronting presents a significant advantage by mitigating the expenses associated with manual detection methods employed in prior research.

-The authors' disclosure regarding the continued support for domain fronting by numerous CDN providers serves as a compelling argument for a reevaluation of the threat posed by domain fronting within the Internet community.

Cons:

-While the authors successfully identify instances of domain fronting through measurement, the comprehensive disclosure of associated vulnerabilities is currently incomplete. Ethical concerns arise as the authors shared their results with only two CDNs and received responses from just one provider, potentially exposing a risk of malicious exploitation by attackers leveraging the insights presented in this paper.

-A more thorough analysis of the risks associated with domain fronting is warranted. This includes delving into aspects such as gang clustering, realistic attack behavior, and other pertinent factors to provide a more comprehensive understanding of the potential hazards associated with this phenomenon.

**Questions:**

-How do the authors rectify the inadequacy in disclosing vulnerabilities within their papers?

-To what extent do correlations exist between malicious domains, and what tangible harm, if any, do they inflict?

-What notable distinctions in approach and outcomes emerge when comparing the current work with previous studies, such as "Assessing and Exploiting Domain Name Misinformation"?

-In what manner do the authors evaluate the efficacy of their autodiscovery methods?

**Ethics Review Description:**

The authors find that 22 CDN providers with domain fronting security vulnerabilities. However, they only disclosed to two providers and received only one response.

**Ethics Review Flag:**

Yes

**Reviewer Confidence:**

4: The reviewer is certain that the evaluation is correct and very familiar with the relevant literature

**Scope:**

3: The work is somewhat relevant to the Web and to the track, and is of narrow interest to a sub-community

---

### Official Review · Reviewer_Qqq3 · 2023-11-24

**Novelty:** 6
**Technical Quality:** 6

**Review:**

The paper describes a systematic approach to discover CDNs that are prone to domain fronting. The proposed approach is based on DNS traffic analysis that first identifies the domains served by CDNs and subsequently discover CDNs prone to domain fronting. A study of major CDNs is provided that highlights that domain fronting is still feasible in a large majority (22 out of 30) of current CDNs.

Pros:
- Simple, yet effective, technique to automate DNS traffic analysis for evaluating CDNs
- Good evaluation (with results of security concern)
- Well written paper with a good description of measurement methodology and experimental evaluation
- Use of responsible disclosures

Cons:
- Lack of comparison with concurrent study
- Responsible disclose process is not completed

The paper is well written with a well-described problem and methodology that is nicely supported by a good evaluation. The overall approach of using DNS analysis, instead of creating new domains, is simple yet effective. The results are concerning as it shows that a majority of CDNs continue to support domain fronting even after the issue has been known for a while. The reviewer liked that the authors have disclosed their findings to the effected CDNs, though would have liked such disclosures to be sent to all of them (if the paper is accepted, this process should be finished before the paper is published).

The study using Virus Total to highlight that 31% of domain fronting CDNs served content from malicious domains is concerned and exemplifies the value of such a study. It would be useful to include the analysis if such domains were reachable using domain fronting (and no filtering is done by the CDN).

While the paper includes a discussion on a concurrent study that is similar to the one described in the paper, it does not provide any arguments on what, if anything, is different in that study even though it does not reduce the overall contributions of the paper.

Post Rebuttal:
I thank the authors for their response and for disclosing the issues with the corresponding CDNs. If the paper is accepted, please update the paper with appropriate responses from the CDNs.
While the reviewer understand that related study was done concurrently, it would be useful to include the explanation provided in the rebuttal in the paper itself.
Overall, the reviewer is satisfied with the response.

**Questions:**

- Why were responsible disclosures not sent to all CDNs? This should be done before the paper is published.
- Can you please compare with the concurrent study?

**Reviewer Confidence:**

3: The reviewer is confident but not certain that the evaluation is correct

**Scope:**

4: The work is relevant to the Web and to the track, and is of broad interest to the community

---

### Decision · Program_Chairs · 2024-01-22

**Decision:**

Accept (Oral)

**Comment:**

### Meta Review:

 **Pros:**
 1. **Insightful Measurements:** The overall consensus is that the measurement study provides valuable insights into the prevalence of domain fronting across Content Delivery Networks (CDNs).
 2. **Ethical Considerations:** The inclusion of responsible disclosures to CDNs is positively acknowledged, demonstrating ethical considerations in the research process.
 3. **Clarity and Presentation:** The paper is commended for its clarity in presentation, making it easy to follow the methodology and results.
 4. **Real-World Impact:** The findings are recognized for their potential real-world impact, especially in making CDNs aware of domain fronting issues.

 **Cons:**
 1. **Automated Testing Concerns:** Some reviewers express concerns about potential false positives/negatives in automated testing and suggest further validation.
 2. **Novelty Questioned:** One reviewer notes that the topic of domain fronting is not particularly new, and the novelty of the contribution may be weakened.
 3. **Lack of In-Depth Analysis:** The absence of a detailed discussion on countermeasures against domain fronting is highlighted as a weakness.
 4. **Comparison with Concurrent Study:** A concurrent study is mentioned, and there is a request for a more explicit comparison to understand differences or unique contributions.

 **Suggestions:**
 1. **Validation and Clarification:** Address concerns about potential false positives/negatives and provide further validation of the automated testing approach.
 2. **Novelty Reinforcement:** Strengthen the novelty of the contribution, possibly by emphasizing unique aspects or differences from prior works.
 3. **In-Depth Analysis:** Include a more comprehensive discussion of plausible countermeasures against domain fronting, enhancing the paper's overall depth.
 4. **Complete Responsible Disclosures:** Ensure responsible disclosures are completed for all CDNs before publication.
 5. **Comparison Clarification:** Explicitly compare the current work with the concurrent study, highlighting any distinctions or unique contributions.

 ### Conclusion:
 The paper receives positive feedback for its contributions to understanding domain fronting vulnerabilities in CDNs. To enhance the paper, reviewers recommend addressing concerns about automated testing, reinforcing the novelty, providing in-depth countermeasure analysis, completing responsible disclosures, and clarifying the comparison with a concurrent study. The ethical considerations and potential real-world impact contribute to the paper's overall positive reception.

 ---